# Comment on Zamfir et al. Hematologic Malignancies Diagnosed in the Context of the mRNA COVID-19 Vaccination Campaign: A Report of Two Cases. *Medicina* 2022, *58*, 874

**DOI:** 10.3390/medicina58111575

**Published:** 2022-11-01

**Authors:** Olgu Erkin Çınar, Batuhan Erdoğdu, Mine Karadeniz, Sercan Ünal, Ümit Yavuz Malkan, Hakan Göker, İbrahim Celalettin Haznedaroğlu

**Affiliations:** Division of Hematology, Department of Internal Medicine, Hacettepe University, 06800 Ankara, Turkey

**Keywords:** BNT162b2, COVID-19, leukemia, malignancy, hematologic adverse event

## Abstract

The SARS-CoV-2 spike protein mRNA-based vaccines have prevented countless mortality and morbidity, and have an excellent risk/benefit ratio. However, various adverse events may rarely occur after the BNT162b2 vaccine, like any other medical intervention. The COVID-19 itself and the spike protein produced endogenously by mRNA vaccines may have immunological, microenvironmental, prothrombotic, and neoplastic effects. As a contribution to the published report, we would like to share our experience regarding four cases in which myeloid neoplasms emerged following the vaccination. *Conclusions*: There is no doubt that vaccination could continue along the lines of established universal recommendations. Meanwhile, all hematological adverse events must be closely monitored and reported. Further efforts should be focused on the probable pathobiological mechanisms and causalities of spike protein-related toxicity and clonal myeloid disorders.

We have read with great interest the outstanding reports on hematological adverse events associated with COVID-19 vaccination by Zamfir et al [1]. The authors brilliantly pointed out the excellent risk/benefit ratio of SARS-CoV-2 spike protein mRNA-based vaccines, as well as indicating the possible hematological neoplastic adverse events. 

We would like to share our experience regarding four patients concerning the emergence of myeloid neoplastic disorders following the BNT162b2 mRNA-based COVID-19 vaccine administration. Herein, we summarize four cases followed in our clinic.


*Case 1*


A 61-year-old man with insignificant medical history applied for a cough complaint on the 30th day after the third dose of the BNT162b2 vaccine. The COVID-19 polymerase chain reaction (PCR) test resulted positive, and a pneumonia diagnosis was made. The patient had also pancytopenia. Almost all the leukocytes were blasts in the peripheral blood smear. The bone marrow biopsy and aspiration studies confirmed the acute myeloid leukemia (AML) diagnosis with 80% blastic infiltration. The NPM1 mutation was detected, and the patient was treated with a classical “3 + 7” regimen for induction. Complete remission was achieved. The subsequent treatment and follow-up continue. 


*Case 2*


An otherwise healthy 28-year-old female without any COVID-19 medical history admitted to the clinic with weakness, oral bleeding, and petechiae in lower extremities. The complaints emerged approximately four weeks after the second dose of the BNT162b2 vaccine. She had leukocytosis and bicytopenia with monoblastic cells in the smear. The bone marrow studies revealed AML with diffuse blastic infiltration, then the “3 + 7” induction was started. After remission was achieved, allogeneic hematopoietic stem cell transplantation (HSCT) from a full-matched sibling took place. She is being followed-up after HSCT in complete remission for a year.


*Case 3*


A 72-year-old man with a history of type II diabetes, hypertension, and coronary artery disease was evaluated for the complaint of black-colored stools. In first-step tests, pancytopenia with profound thrombocytopenia (9 × 10^3^/μL) was detected as the cause of melena originating from the upper gastrointestinal bleeding. The patient had been given the fifth dose of the BNT162b2 vaccine approximately five weeks before the complaints emerged. The bone marrow aspiration and biopsy demonstrated 70% blastic infiltration and an AML diagnosis was made. The combination of venetoclax and azacytidine has initiated.


*Case 4*


A 60-year-old man with no known chronic disease and COVID-19 medical history applied to another center due to the fast-expanding swellings on the vertex, neck, and left armpit. He was referred to our clinic for the vertical lesions (Figure 1) and lymphadenopathies on physical examination. The patient had received the fourth shot of the BNT162b2 vaccine a month before the complaints started. He had no B symptoms, and the complete blood count was normal. The vertical lesion was sampled, and the pathologic evaluation resulted in granulocytic sarcoma of CD34, CD123, and MPO positive immature cells. A PET-CT imaging showed increased FDG uptake on the vertical lesion and occipital region, as well as on the axillary, cervical, mediastinal, abdominal, and inguinal lymph nodes. The bone marrow biopsy and aspiration studies showed a 4–5% blastic cells percentage, and the sarcoma and lympadenomegalies were not in relation to AML. The isolated extramedullary myeloid sarcoma diagnosis was made, but a leukocytosis developed within days, before treatment was even started. The bone marrow studies were repeated and 30% myeloid blasts were seen, eventually the patient was diagnosed with AML. Due to the rapidly deteriorating clinical condition, the venetoclax–azacytidine combination was started for remission induction. After the first four-week course, physical examination findings completely disappeared. The complete medullary remission was achieved. After the second course, the PET-CT imaging showed no extramedullary metabolic activity. 

We had reported a case series with hematological adverse events after mRNA vaccines [2]. Here, we wanted to discuss the possible relationship of BNT162b2 with myeloid leukemogenesis in the context of pharmacovigilance through four cases. Because the cases were generally healthy and did not have frequent hospital admissions, pre-vaccine test results were not available in their health records. HSCT candidate patients were treated with the classic "3+7" regimen, while elderly or comorbid patients were treated with the modern molecular approach, the venetoclax-azacytidine regimen [3].

Among the adverse events, cytopenias are much more common, and various malignancies were present other than myeloid neoplasms. There are many possible reasons and postulates behind these adverse events. In a recent review [4], we also summarized some concepts in terms of immunological, microenvironmental, prothrombotic, and neoplastic aspects of the SARS-CoV-2 virus infection itself as well as the spike protein produced endogenously by mRNA vaccines.

We add our voice to Zamfir and coworkers to state that COVID-19 vaccines have saved countless lives and helped to save humanity from the devastating outcomes of the SARS-CoV-2 virus. There is no doubt that vaccination could continue within the line of established universal recommendations. However, this should not distract health professionals from basic pharmacovigilance principles and reporting adverse events. 

Above all, one should not fall into the post hoc ergo propter hoc fallacy, and it is not possible to directly link these adverse effects to vaccines unless there is strong evidence of causality. Objective pharmacovigilance-based follow up is needed to establish a clinicobiological correlation between spike-mRNA vaccines and leukemogenesis. Meanwhile, all hematological adverse events must be closely monitored and reported. Further efforts should be focused on the probable pathobiological mechanisms and causalities of spike protein-related toxicity and clonal myeloid disorders. 

## Figures and Tables

**Figure 1 medicina-58-01575-f001:**
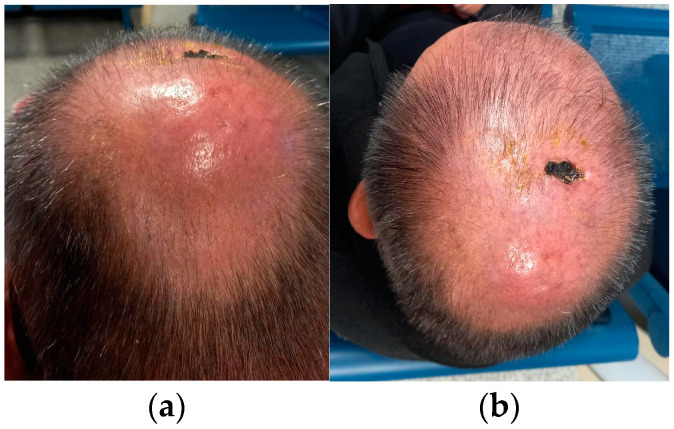
The vertical extramedullary tumor with necrotic appearance and the parietooccipital erythematous lesion of the patient with granulocytic sarcoma. (**a**) The back view shows erythematous swelling; (**b**) The view from the top; dried exudate and necrotic ulcer-like appearance of the sarcoma.

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
