# Peer review of "Comment on Zamfir et al. Hematologic Malignancies Diagnosed in the Context of the mRNA COVID-19 Vaccination Campaign: A Report of Two Cases. Medicina 2022, 58, 874"

_medicina, 2022, doi:10.3390/medicina58111575_

Round 1

Reviewer 1 Report

In the four reported cases, there is only a temporal relationship between the anti-Covid vaccination and the occurrence of acute myeloid leukemia. The Authors should make an effort to better support a possible causal role between the two events.

Author Response

The following answers were given and the revisions were made according to the reviewers’ comments:

Answer:

Although the onset of symptoms is reported to be shorter (a few days) after the doses of the mRNA vaccines in the reported cases, symptom onset may be different for myeloid malignancies in our patients. Evidence-based answers are difficult as no objectively established time to symptom onset has been previously reported for possible leukemia development after mRNA vaccines. On the other hand, when we apply the Naranjo adverse event probability scale, we get the result of "possible adverse event" with a score of 2.

We would like to thank the Reviewer for the comments.

We totally agree with the reviewer and we have corrected the grammar and spelling mistakes as recommended.

Reviewer 2 Report

In this Comment to the recent report by Zamfir et al, the authors describe the emergence of four very interesting cases of acute myeloid leukemia/granulocytic sarcoma temporally arising after administration of the BNT162b2 mRNA-based COVID-19 vaccine. The authors correctly conclude that it is not possible to directly link these adverse events to vaccines. However, the authors rightfully suggest to pursue a stringent pharmacovigilance methodology that includes also reporting of hematological malignancies.

MAJOR ISSUES
1. Was a complete blood count available for these patients before receiving the BNT162b2 mRNA-based COVID-19 vaccine dose temporally related to the diagnosis of acute myeloid leukemia/granulocytic sarcoma? In case, did the complete blood count at that time reveal any abnormality that might be suggestive of leukemia?

2. Two patients were treated with 3+7 (followed by HSCT in one) and two patients received venetoclax + azacytidine. I suggest that the authors include one short sentence on the innovations in therapeutic landscape of acute myeloid leukemia, referring to the very recent review by Gallazzi et al. Int J Mol Sci. 2022 Jul 7;23(14):7542. doi: 10.3390/ijms23147542

MINOR ISSUES

1.     In the abstract (lines 14-15), the sentence “…we would like to share our experience regarding the cases in which myeloid neoplasms emerged…” should read ““…we would like to share our experience regarding four cases in which myeloid neoplasms emerged…”

Author Response

Answer 1

Unfortunately, pre-vaccine test results could not be found, as the patients were healthy individuals who had no previous test indication. Therefore, it was not possible to determine precisely when the leukemic preclinical phase began.

We would like to thank Reviewer for the legitimate question.

Answer 2

We would like to express our gratitude to the Reviewer for these comments that help us to improve our manuscript. The citation was added to the manuscript in the relevant section.

Answer 3

Thank you very much to the reviewer for the correction. The relevant change has been made.

Round 2

Reviewer 1 Report

The revised version of the paper is clearer and acceptable for publication.

Reviewer 2 Report

The authors have addressed all the issues that had been raised. No further comments from my side.